# Mass Spectrometric Characterization of Epoxidized Vegetable Oils

**DOI:** 10.3390/polym11030394

**Published:** 2019-02-28

**Authors:** Ákos Kuki, Tibor Nagy, Mahir Hashimov, Stella File, Miklós Nagy, Miklós Zsuga, Sándor Kéki

**Affiliations:** Department of Applied Chemistry, Faculty of Science and Technology, University of Debrecen, Egyetem tér 1., H-4032 Debrecen, Hungary; kuki.akos@science.unideb.hu (Á.K.); nagy.tibor@science.unideb.hu (T.N.); mahir.hashimov@yandex.com (M.H.); file.stella96@gmail.com (S.F.); miklos.nagy@science.unideb.hu (M.N.); zsuga.miklos@science.unideb.hu (M.Z.)

**Keywords:** mass spectrometry, mass-remainder analysis, vegetable oils, triglycerides, epoxidation, biopolymers

## Abstract

Matrix-assisted laser desorption ionization and electrospray ionization mass spectrometry (MALDI-MS and ESI-MS) were used for the characterization of epoxidized soybean and linseed oils, which are important raw materials in the biopolymer production. The recently invented data mining approach, mass-remainder analysis (MARA), was implemented for the analysis of these types of complex natural systems. Different epoxidized triglyceride mass spectral peak series were identified, and the number of carbon atoms and epoxide groups was determined. The fragmentation mechanisms of the epoxidized triglyceride (ETG) adducts formed with different cations (such as H^+^, Na^+^, Li^+^, and NH_4_^+^) were explored. As a novel approach, the evaluation of the clear fragmentation pathways of the sodiated ETG adducts enabled the estimation of the epoxidized fatty acid compositions of these types of oils by MS/MS.

## 1. Introduction

Recently, there has been an increased demand for the preparation of polymeric materials based on renewable sources driven by environmental concerns and the depletion of fossil resources. Vegetable oils, such as soybean, sunflower and linseed oils, are one of the most important sources of biopolymers [1,2,3,4,5,6,7]. Their main constituents are triglycerides, i.e., triesters of glycerol with saturated and unsaturated fatty acids (FAs). There is a wide variety of triglyceride oil-based polymers, for instance, oxypolymerized oils, polyesters, polyurethanes, polyamides, epoxy resins, etc. [3]. Moreover, the use of photopolymerization, considered to be a green technology, can further decrease the environmental load of biopolymer production [8,9]. Double bonds in the unsaturated fatty acids of the triglycerides can be converted to the more reactive epoxide functional groups to facilitate the polymerization process [10,11,12,13]. 

Since vegetable oils vary widely in their chemical microstructure, the choice of triglyceride oil has a crucial influence on the properties of the polymer product. Therefore, the characterization of the triglyceride oils is essential for the design of new materials and for quality control in biopolymer production. Characterization usually means the determination of the fatty acid composition, the degree of unsaturation, the number of epoxide groups per molecule (NEG) and the degree of epoxidation (DOE, the percentage of conversion from double bonds to epoxide groups) of the epoxidized triglyceride oils. The most widely used methods to characterize triglyceride oils are infrared spectroscopy, particularly Fourier transform infrared (FTIR) spectroscopy [14], gas chromatography–mass spectrometry (GC-MS) [15], nuclear magnetic resonance (NMR) spectroscopy [14] and electrospray ionization mass spectrometry (ESI-MS) [16]. However, to the best of our knowledge, there have been no studies involving the mass spectrometric characterization of the epoxidized triglyceride oils, which are important raw materials in biopolymer production. Accordingly, the main aim of our work was the evaluation of the chemical structure of epoxidized soybean and linseed oils using soft ionization tandem mass spectrometry (MS/MS).

## 2. Materials and Methods

### 2.1. Chemicals

The epoxidized triglyceride oils were produced by Arkema (Colombes, France). The samples were dissolved in HPLC-MS grade methanol purchased from VWR International (Leuven, Belgium).

### 2.2. Matrix-Assisted Laser Desorption/Ionization Time-of-Flight Mass Spectrometry (MALDI-TOF MS)

The MALDI-TOF MS measurements were carried out with a Bruker Autoflex Speed mass spectrometer (Bruker Daltoniks, Bremen, Germany) equipped with a time-of-flight (TOF) mass analyzer. In all cases, 19 kV (ion source voltage 1) and 16.65 kV (ion source voltage 2) were used. For reflectron mode, 21 kV and 9.55 kV were applied as reflector voltage 1 and reflector voltage 2, respectively. A solid phase laser (355 nm, ≥100 μJ/pulse) operating at 500 Hz was applied to produce laser desorption and 15,000 shots were summed. The MALDI-TOF MS spectra were internally calibrated with a mixture of α-,β-,γ-cyclodextrin, rutin and lactose octaacetate.

The matrix used for the MALDI-TOF MS was 2′,4′,6′-trihydroxyacetophenone (THAP), dissolved in methanol at a concentration of 20 mg/mL. The epoxidized oils were also dissolved in methanol at a concentration of 10 mg/mL. Sodium trifluoroacetate was used as the ionizing agent (5 mg/mL). The mixing ratio was 50/10/5/2 (matrix/oil/cationizing agent/internal standard mixture). A volume of 0.25 μL of the solution was deposited onto a metal sample plate and allowed to air-dry.

### 2.3. Electrospray Quadrupole Time-of-Flight Mass Spectrometry (ESI-QTOF MS)

A Maxis II type Qq-TOF MS instrument (Bruker Daltoniks, Bremen, Germany) equipped with an Apollo II electrospray ion source was used. The spray voltage was 4.5 kV. The resolution of the instrument was 40,000 at *m*/*z* 400 (FWHM), and the mass accuracy was <2 ppm (external calibration). N_2_ was utilized as the drying gas (200 °C, 4.0 L/min), nebulizer gas (0.5 bar) and collision gas. The collision voltage was varied in the range of 20–90 V. The mass spectra were recorded by means of a digitizer at a sampling rate of 2 GHz. The spectra were calibrated externally by ESI tune mix, from Bruker. The spectra were evaluated with the Compass Data Analysis 4.4 software from Bruker (Bremen, Germany). The sample solutions were introduced directly into the ESI source with a syringe pump (Cole-Parmer Ins. Co., Vernon Hills, IL, USA) at a flow rate of 3 μL/min. The concentration of the samples was 0.01 mg/mL.

## 3. Results and Discussion

The MALDI-MS spectra of the epoxidized soybean and linseed oils are shown in Figure 1.

As seen in Figure 1, the sodiated adducts of the epoxidized triglycerides (ETGs) were detected in the mass range of *m*/*z* 880 to 1040. The blue labels above the peaks indicate the number of carbon atoms and epoxide groups. For example, the (55:2) peak at *m*/*z* 913.746 corresponds to the [C_55_H_102_O_8_ + Na]^+^ adduct ion with a theoretical *m*/*z* 913.7467. The number of carbon atoms and epoxide groups of this ion reveal that the chemical formula of this ETG was C_3_H_8_[CO_2_(CH_2_)_15_(CHOCH)_1_]_2_[CO_2_(CH_2_)_15_] or C_3_H_8_[CO_2_(CH_2_)_17_][CO_2_(CH_2_)_13_(CHOCH)_2_][CO_2_(CH_2_)_15_], with the epoxidized FA composition of oleic, oleic, palmitic or stearic, linoleic, palmitic acid, respectively (in the compositions, CO_2_ is the carboxyl group of the fatty acid, and CHOCH is the epoxy group). The mass peak identification, namely the carbon and epoxy number assignment, is not always straightforward, because the replacement of two CHOCH groups by six CH_2_ groups results in a 0.0728 *m*/*z* difference, as was observed for the (55:5) and (57:3) ETGs at *m*/*z* 955. To resolve these closely spaced ions, a resolving power of *m*/Δ*m*_50%_ > 13,700 at *m*/*z* 1000 was required (see Figure 1 insets). Comparing the mass spectra of the soybean and linseed oils, the remarkable differences in the intensities of the corresponding peaks reflect the different triglyceride (and fatty acid) compositions of the two vegetable oils, as will be detailed later.

Despite the simplicity of the mass spectrum shown in Figure 1, the vegetable triglyceride oils and their epoxidized forms, being samples of natural origin, are complex systems with numerous minor intensity mass peaks with a wide variety in the numbers of double bonds (or epoxide groups) and CH_2_ groups. Furthermore, the side products of the epoxidation can also be observed, e.g., diols are formed by the subsequent addition of water. Hence, the mass spectrometric study of complex natural samples requires effective data processing and visualization methods [17]. Recently, we proposed a simple algorithm, mass-remainder analysis (MARA), for the processing of complex copolymer mass spectra [18]. In the first step, MARA assigns a mass remainder (MR) value to the mass spectrum peaks defined by Equation (1),
*MR* = *m*/*z* MOD *B*,(1)
where *B* is the exact mass of a base unit and the modulo (MOD) operation finds the remainder after the division. For example, the MR value of the most intense peak in Figure 1a at *m*/*z* 997.696 is 2.5848 (see the largest dot in Figure 2b), because 997.696 = 71 × 14.01565 + 2.5848, where 14.01565 is the exact mass of the base unit *B*, such as CH_2_. Choosing the CH_2_ group as the base unit, the mass peaks of the ETGs can be easily identified and the number of epoxide groups (NEG) can be simultaneously determined, as detailed in the following.

Figure 2 depicts the MR versus *m*/*z* plot of the MALDI mass spectrum of the epoxidized soybean oil shown in Figure 1. In the case of the total (100%) conversion of double bonds to epoxide groups (DOE = 100%), the MR values depended merely on the NEG. An additional epoxide ring, i.e., the replacement of a (CH_2_)_2_ moiety by a C_2_H_2_O group, resulted in a −0.0364 shift in the MR value, as illustrated by the horizontal lines in Figure 2b. (Note that the compounds differing only in the number of CH_2_ groups have the same MR values.) Using NEG–MR mapping, the ETGs with no remaining double bonds (DOE = 100%) can be identified in the complex mass spectra, and the number of epoxide groups and subsequently the number of carbon atoms in the ETG can be determined. For example, an MR = 2.7288 value was calculated for the peak at *m*/*z* 913.746, meaning that NEG = 2 (see Figure 1 and Figure 2b). Table 1 summarizes the identified epoxidized triglycerides along with their relative intensities.

In our experiments, the 100% conversion of double bonds to epoxide rings was typical (see Figure 2a red highlight, and Figure 2b). Nevertheless, other minor intensity series were also identified or filtered by means of MARA. As seen in Figure 2a, the triglycerides with one remaining double bond (yellow highlight) and diol side products (blue highlight) were also identified. The identification of these series in the mass spectrum was not straightforward, but MARA was able to filter these compound classes based on their MR values. In addition, this grouping can be visualized in the MR versus *m*/*z* plot, as seen in Figure 2. The rest of the mass peaks, which did not belong to any identified series, were mostly isotope peaks (see Figure 2a, light blue dots). As an advantage of the MARA data processing method, the isotopes were clearly separated exactly by the ΔMR = 1 mass remainder value difference.

Once the epoxidized triglyceride oil mass peaks have been filtered and the chemical compositions (i.e., the carbon and epoxy numbers) assigned to the peaks, the usual polymer quantities can be accurately calculated. Table 2 lists the number average molecular weight *M_n_*, the average number (*n_n_^NEG^*) and average number of epoxide groups weighted by number of epoxide groups (*n_w_^NEG^*), the polydispersity of the number of epoxide groups (*n_n_^NE*^*/*n_w_^NEG^*), average number of carbon atoms (*n_n_^C^*), and the degree of epoxidation (*DOE*). The *M_n_*, *n_n_^NEG^*, *n_w_^NEG^* and *n_n_^C^* were calculated using the following equations:(2)$$M_{n}=\;\frac{\sum_{i=1}^{\infty }\left(I_{i}\times m_{i}\right)}{\sum_{i=1}^{\infty }I_{i}}$$
(3)$$n_{n}^{NEG}=\;\frac{\sum_{i=1}^{\infty }\left(I_{i}\times n_{i}^{NEG}\right)}{\sum_{i=1}^{\infty }I_{i}}$$
(4)$$n_{w}^{NEG}=\;\frac{\sum_{i=1}^{\infty }\left(I_{i}\times n_{i}^{NEG}{}^{2}\right)}{\sum_{i=1}^{\infty }\left(I_{i}\times n_{i}^{NEG}\right)}$$
(5)$$n_{n}^{C}=\;\frac{\sum_{i=1}^{\infty }\left(I_{i}\times n_{i}^{C}\right)}{\sum_{i=1}^{\infty }I_{i}}$$
where *I_i_* is the intensity of the peak of interest, *m_i_* is the molecular weight of the ETGs, and *n_i_^NEG^* is the number of epoxide groups in a molecule.

The knowledge of these quantities, for example the average number of epoxide groups per molecules, is essential in the design and production of vegetable oil-based biopolymers.

Our next goal was to gain a deeper insight into the structure of the epoxidized triglycerides of natural origin by soft ionization tandem mass spectrometry. In the following, we will use the epoxidized prefix to indicate that all the double bonds have been converted to epoxide groups in the molecule. In the first step, the fragmentation mechanisms of the ETG adducts formed with different cations (such as H^+^, Na^+^, Li^+^, NH_4_^+^) were explored. The ESI-QTOF MS/MS spectra of the linseed and soybean oils are presented in Figure 3, and the sodiated ETG adduct at *m*/*z* 997 (57:6) was selected as the precursor ion for the MS/MS. Additional MS/MS examples for the ammoniated and lithiated adducts are shown in the Appendix A as Appendix A, respectively.

It seems at first glance, that the product ion spectra of the sodiated adduct ions are simpler than those of other adducts formed with H^+^, Li^+^, and NH_4_^+^ ions. As seen in Figure 3, the significant reaction in the collision-induced dissociation (CID) process is the decomposition of the sodiated ETG to yield an epoxidized diglyceride and a single epoxidized fatty acid (EFA). In contrast, various other reactions can be observed, resulting in more complex product ion spectra for the lithiated and ammoniated adducts (see Appendix A), including water elimination and EFA backbone cleavage. The proposed fragmentation pathways for the sodiated and ammoniated ETG adducts are presented in the Appendix A as Appendix A, respectively. It can also be observed, that even though the same sodiated precursor ions, such as *m*/*z* 997 (57:6), were selected, the intensity ratios of the product ions showed large differences in the case of epoxidized linseed and soybean oil (see Figure 3a,b). For example, the intensity ratios of the product ions at *m*/*z* 335 and 349 (corresponding to the sodiated adducts of the epoxidized linoleic acid (ELA) and epoxidized linolenic acid (ELNA), respectively) were 0.32 and 4.4 for the linseed and soybean oil, respectively. This huge difference is in line with the different LA and LNA content of the two vegetable oils (see Figure 4). Therefore, the clear product ion spectra of the sodiated ETG adducts provide a possibility to determine the EFA compositions of these oils by MS/MS, as a novel approach. The breakdown curves, i.e., the relative intensities of product ions versus collision energy plot of ETGs suggest that below a collision energy of about 70 eV the simultaneous reactions were dominant in the product ion formation. Thus, the consecutive reactions can be neglected, since the breakdown curves of the product ions with two EFA chains start to decrease above 70 eV (see the breakdown plot of the sodiated ETG (57:6) as an example in Appendix A). It means that the fragmentation reactions can be simplified as the loss of an EFA chain. For instance, if the lost EFA is the epoxidized linolenic acid (ELNA), the sodium ion can remain attached either to the epoxidized diglyceride, resulting in the product ion at *m*/*z* 671 in Figure 3, or to the single EFA (ELNA in this case) yielding the product ion at *m*/*z* 349. Furthermore, this approach enables the estimation of the percentage of ELNA in the ETG (57:6) specimen (the precursor ion at *m*/*z* 997 in Figure 3), as the relative intensity of the sum of the product ions at *m*/*z* 671 and 349 is related to the sum of all the product ions. Performing this calculation for all the ETG specimens (after recording their MS/MS spectra), and summarizing the percentages of the EFA weighted by the corresponding intensities of the ETG specimens in the MS spectrum (see Figure 1b in the case of linseed oil), provides the percentage of the EFA in the epoxidized triglyceride oil. Figure 4 shows the epoxidized fatty acid composition of the epoxidized linseed oil calculated by our approach compared to the theoretical values found in the literature [3] (the EFA composition of soybean oil is presented in Appendix A). The similarity of the EFA distributions suggests that our simple method is capable of estimating the epoxidized fatty acid composition of the epoxidized triglyceride oils, and that it does not require any complicated and time-consuming sample preparation, derivatization, or separation prior to MS measurements.

## 4. Conclusions

This study explored the applicability of soft ionization mass spectrometry, including MALDI and ESI, for epoxidized vegetable triglyceride oils, which are important polymer raw materials. Mass-remainder analysis (MARA), our recently invented data mining procedure, was used to process the mass spectra of these complex samples of natural origin. MARA was particularly useful for the identification of different epoxidized triglyceride series, such as triglycerides with 100% DOE (with all the double bonds converted to epoxide groups), triglycerides with one remaining double bond, or with diol side products. Furthermore, MARA facilitated the determination of the number of epoxide groups and subsequently the number of carbon atoms in the triglyceride. In addition, our MS/MS experiments revealed that the sodiated ETG adduct ions have very clear fragmentation pathways, in contrast to those of the other adducts (H^+^, Li^+^, and NH_4_^+^). This important finding enables the estimation of the epoxidized fatty acid composition of epoxidized triglyceride oils by means of MS/MS, without any complicated and time-consuming sample preparation, derivatization, or separation. It can be particularly useful in the characterization of the fatty acids variety of complex vegetable oil mixtures. Interestingly, this clear and informative fragmentation behavior does not apply to the sodiated adducts of the non-epoxidized triglyceride oils due to their inefficient fragmentation in CID experiments.

## Figures and Tables

**Figure 1 polymers-11-00394-f001:**
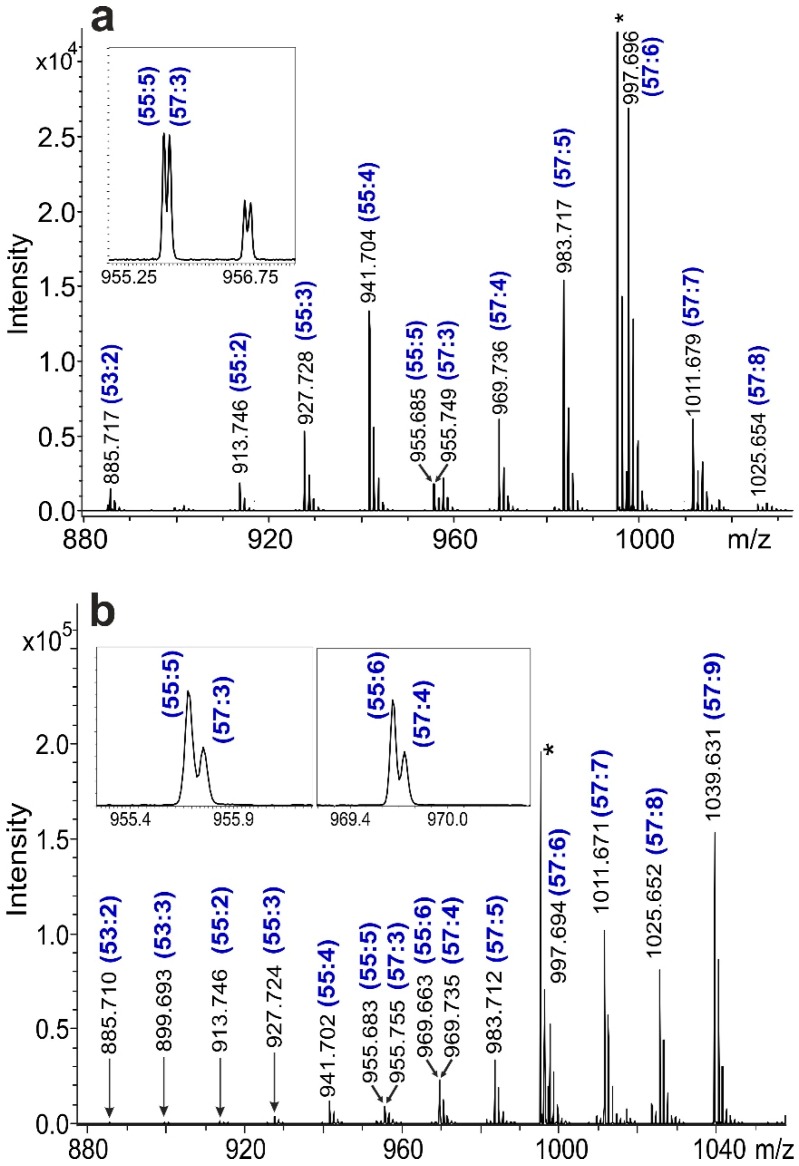
MALDI-TOF mass spectrum of the (**a**) epoxidized soybean oil and (**b**) epoxidized linseed oil. The blue labels above the peaks indicate the number of carbon atoms and epoxide groups. The asterisk denotes the internal calibrant.

**Figure 2 polymers-11-00394-f002:**
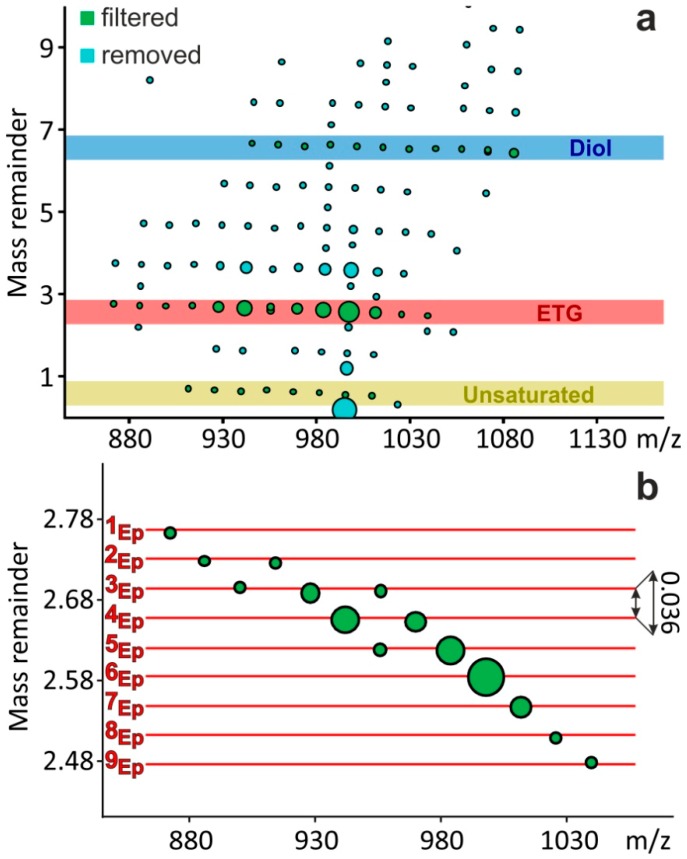
(**a**) Mass-remainder (MR) versus *m*/*z* plot, and (**b**) zoomed MR versus *m*/*z* plot of the MALDI-TOF mass spectrum of the epoxidized soybean oil. ETG stands for the epoxidized triglycerides with no remaining double bonds. The size of the larger dots indicates peak intensity. For the sake of visibility, a minimum dot-size was used, therefore the sizes of the unsaturated and diol dots are exaggerated, and do not reflect the real intensity ratios.

**Figure 3 polymers-11-00394-f003:**
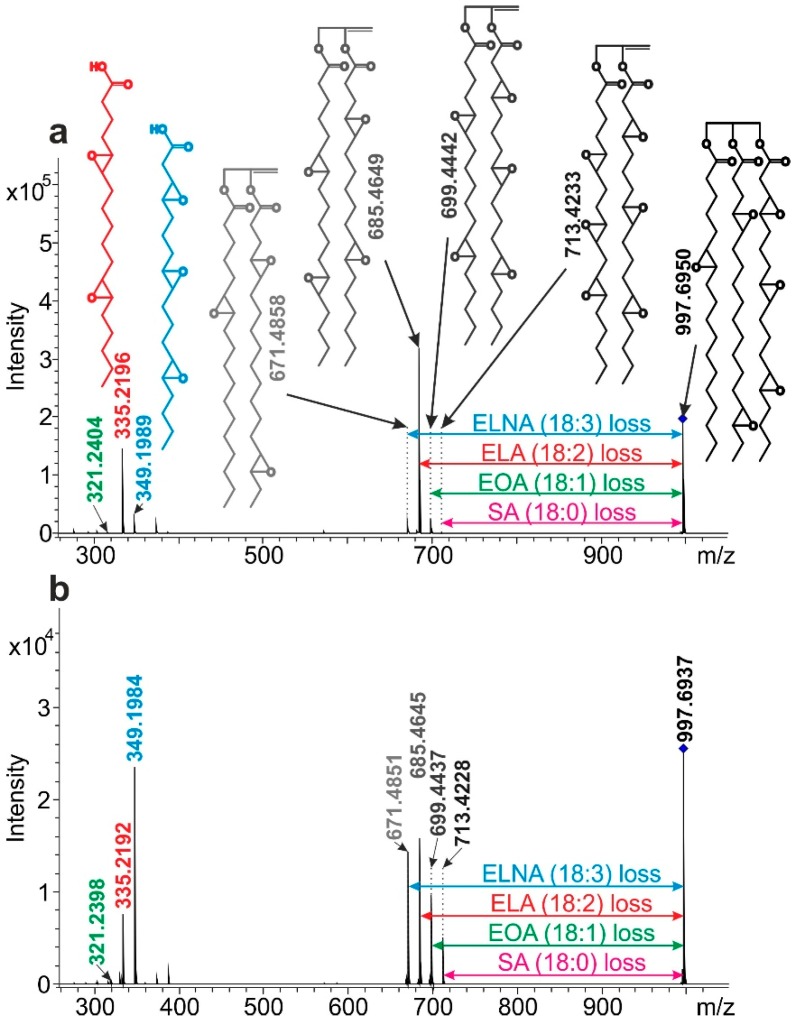
MS/MS spectra of the sodiated (57:6) ETG adduct at *m*/*z* 997 of the (**a**) linseed and (**b**) soybean oil recorded at a laboratory frame collision energy of 70 eV. Abbreviations: SA, stearic acid (18:0); EOA, epoxidized oleic acid, (18:1); ELA, epoxidized linoleic acid (18:2); ELNA, epoxidized linolenic acid (18:3). The chemical structure of the precursor ion (*m*/*z* 997) is just an example, it can also contain, for example, three ELA (18:2) chains.

**Figure 4 polymers-11-00394-f004:**
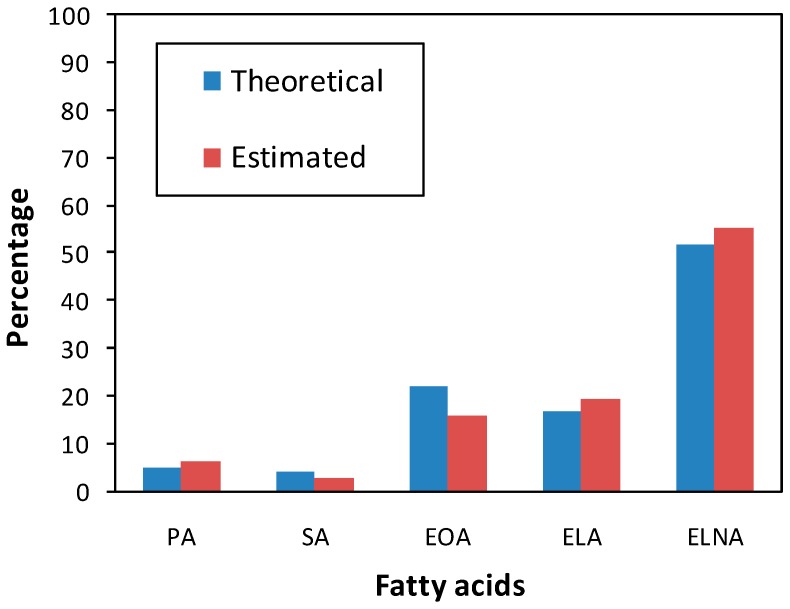
Fatty acid composition of the epoxidized linseed oil calculated by our approach compared to the theoretical values (see [3]). Abbreviations: PA, palmitic acid, (16:0); SA, stearic acid (18:0); EOA, epoxidized oleic acid, (18:1); ELA, epoxidized linoleic acid (18:2); ELNA, epoxidized linolenic acid (18:3).

**Table 1 polymers-11-00394-t001:** The relative intensities of the identified saturated epoxidized oil components determined by MALDI-TOF MS. The epoxidized triglycerides (ETG) compounds are defined by their number of carbon atoms and epoxide groups.

ETG Compound	Soybean Oil (%)	Linseed Oil (%)
(53:2)	2.7	0.1
(53:3)	---	0.1
(55:2)	2.7	0.4
(55:3)	17.3	0.7
(55:4)	3.9	2.4
(55:5)	3.4	1.9
(55:6)	---	1.0
(57:3)	9.2	4.7
(57:4)	18.0	2.4
(57:5)	31.3	6.9
(57:6)	10.1	10.7
(57:7)	1.2	20.8
(57:8)	0.1	16.6
(57:9)	---	31.4

**Table 2 polymers-11-00394-t002:** Characterization of the epoxidized soybean and linseed oils.

Property	Soybean Oil	Linseed Oil
*M_n_*	954	989
*n_n_^NEG^* *	5.12	7.23
*n_w_^NEG^* *	5.45	7.60
*n_n_^NEG^* */*n_w_^NEG^* *	1.06	1.05
*n_n_^C^*	56.4	56.8
*DOE* **	99.1	99.1

* *NEG* stands for the number of epoxide groups; ** *DOE* stands for the degree of epoxidation.

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
