# Peer review of "Mass Spectrometric Characterization of Epoxidized Vegetable Oils"

_polymers, 2019, doi:10.3390/polym11030394_

Round 1

Reviewer 1 Report

The manuscript polymers-447826 deals with characterization of epoxidized soyabeen and linseed oils by means of MALDI and ESI mass spectrometry and using Mass reminder analysis that was developed by authors in their previous publication. The topic is interesting, significant and fits with the scope of Polymers. I would suggest to publish the paper after taking into consideration the following comments:

Experimental part, Sentence: “Samples for MALDI-TOF MS were prepared with 2′,4′,6′-trihydroxyacetophenone (THAP) matrix dissolved in methanol at a concentration of 20 mg/mL, the epoxidized oils were also dissolved in methanol at a concentration of 10 mg/mL.“

Please modify the sentence – samples were not prepared with THAP. This is the preparation of matrix solution. The sample were prepared by dissolution with ... etc. Two separate sentences, one for matrix preparation and one for sample preparation would be better.

Experimental part, Sentence: “The collision energy was varied in the range of 20-90 eV (laboratory frame). “ What the “laboratory frame“ means. Please explain in manuscript in more details.

Results and Discussion, Figure 1, I would suggest to move the spectrum of the epoxidized linseed oil from Supplementary material to the manuscript (for instance as Figure 1b) and discuss it as the spectrum of  the epoxidized soybean oil.

Figure 1 caption – asterix --> asterisk

Are you sure that no signal in the oils belong to the internal standard mixture? Please control and consider to express it explicitly in Experimental part.

Line 90, the expression of composition “C3H8[CO2(CH2)16(CO)1]2[CO2(CH2)15] or C3H8[CO2(CH2)17][CO2(CH2)15(CO)2][CO2(CH2)15]“ – it is not clear what CO2, (CO)1 means. Please, explain it better. CO2 is usually carboxyl. Epoxy group could be CH(O)CH or another. But at any case, particular parts of the formulas should be somehow explained (denoted) in the text

Also the Equation 1 is not clear enough. Please add more detailed information on how the “modulo” (MOD) is calculated. Maybe an example of the modulo and MR calculation for a given measured mass would help to understand.

Author Response

Dear Reviewer,

Thank you for reviewing our manuscript entitled ‘Mass spectrometric characterization of epoxidized vegetable oils’. Our answers to your comments are as follows.

– Experimental part, Sentence: “Samples for MALDI-TOF MS were prepared with 2′,4′,6′-trihydroxyacetophenone (THAP) matrix dissolved in methanol at a concentration of 20 mg/mL, the epoxidized oils were also dissolved in methanol at a concentration of 10 mg/mL.“

Please modify the sentence – samples were not prepared with THAP. This is the preparation of matrix solution. The sample were prepared by dissolution with ... etc. Two separate sentences, one for matrix preparation and one for sample preparation would be better.

 Reply: It was modified accordingly, as follows:

 “2′,4′,6′-trihydroxyacetophenone (THAP) was used as the matrix for MALDI-TOF MS, dissolved in methanol at a concentration of 20 mg/mL. The epoxidized oils were also dissolved in methanol at a concentration of 10 mg/mL.”

 – Experimental part, Sentence: “The collision energy was varied in the range of 20-90 eV (laboratory frame). “ What the “laboratory frame“ means. Please explain in manuscript in more details.

 Reply: Laboratory frame collision energy was replaced with “collision voltage” to make it more clear, and as suggested by Reviewer 2, as follows:

 “The collision voltage was varied in the range of 20-90 V.”

 – Results and Discussion, Figure 1, I would suggest to move the spectrum of the epoxidized linseed oil from Supplementary material to the manuscript (for instance as Figure 1b) and discuss it as the spectrum of the epoxidized soybean oil.

 Reply: The spectrum of the epoxidized linseed oil was moved from Supplementary material to the manuscript (Fig. 1b) and discussed, as suggested:

 “Comparing the mass spectra of the soybean and linseed oils, the remarkable differences in the intensities of the corresponding peaks reflect the different triglyceride (and fatty acid) compositions of the two vegetable oil, as it will be detailed later.”

Figure 1 caption – asterix --> asterisk

 Reply: It was corrected, accordingly.

Are you sure that no signal in the oils belong to the internal standard mixture? Please control and consider to express it explicitly in Experimental part.

 Reply: The oils were measured without internal standard addition, as well. It was found, that only the mass peak denoted by asterisk belong to the internal calibrant mixture in the m/z range 880 to 1040.

Line 90, the expression of composition “C3H8[CO2(CH2)16(CO)1]2[CO2(CH2)15] or C3H8[CO2(CH2)17][CO2(CH2)15(CO)2][CO2(CH2)15]“ – it is not clear what CO2, (CO)1 means. Please, explain it better. CO2 is usually carboxyl. Epoxy group could be CH(O)CH or another. But at any case, particular parts of the formulas should be somehow explained (denoted) in the text.

 Reply: It was modified, as suggested, and explained in the text, as follows:

 “(In the compositions, CO2 is the carboxyl group of the fatty acid, and CHOCH is the epoxy group.)”

 Also the Equation 1 is not clear enough. Please add more detailed information on how the “modulo” (MOD) is calculated. Maybe an example of the modulo and MR calculation for a given measured mass would help to understand.

 Reply: Equation 1 was explained and more detailed, as follows:

 “For example, the MR value of the most intense peak in Fig. 1a at m/z 997.696 is 2.5848 (see the largest dot in Fig. 2b), because 997.696 = 71 × 14.01565 + 2.5848, where 14.01565 is the exact mass of the base unit B, such as CH2. “

Thank you for your help!

 Yours sincerely,

 Dr. Sándor Kéki

Professor and Head

Reviewer 2 Report

Dear Editor, dear Authors,

the submitted polymers-447826 manuscript “Mass spectrometric characterization of epoxidized vegetable oils” reports on the use of two techniques of mass spectrometry and of a method of data reduction that the Authors recently published to investigate the composition of a fully epoxidized preparation of vegetable oil which is a starting material for the production of polymers from renewable sources. The topic may fall into those of the Polymers MDPI journal and the makeup of the manuscript is adequate for publishing.

A few comments:

A question: how is it sure that the vegetable oil is fully epoxidized? Although as a mass spectrometrist I agree that we strive to use MS alone, is there any independent, such as manufacturer’s, analysis (residual Iodine number, FAME analysis, …)? Also, you don’t see any addition of MeOH to the epoxide in the ESI-MS? If it is so, this is something you may notice and show a source spectrum in the Supplementary, since it demonstrates that your characterization is solid. In fact, in Figure 2 (upper) there are two other rows, one labelled as Unsaturated, the other as Diol, and both are labelled as Filtered (out). From the dot size, I can guess that there is a substantial fraction of unsaturated TGs remaining, and that some has the epoxide ring(s) opened to the diol. Unless this is explained, it is contradictory to the statement of line 131.

As for the reported MS-MS of isomeric sodiated epoxyTGs, it would be convenient to show the breakdown curves of both isomers, since it is not clear if the two panels of Figure 3 refer to the same isomer or to different ones (I would say they come from different TGs).

Figure S6 mistyped “Collision Energy (eV)”; by the way, it is rather “Collision voltage (DV)”, while center-of mass-collision energy (calculated with the well-known equation Ecm = Elab *(mGas/(mGas + mIon), in eV) is more appropriate when the Authors analyze different precursor ions. Are the curves for the other prominent precursors similar?

Best regards

Author Response

Dear Reviewer,

Thank you for reviewing our manuscript entitled ‘Mass spectrometric characterization of epoxidized vegetable oils’. Our answers to your comments are as follows.

– A question: how is it sure that the vegetable oil is fully epoxidized? Although as a mass spectrometrist I agree that we strive to use MS alone, is there any independent, such as manufacturer’s, analysis (residual Iodine number, FAME analysis, …)?

 Reply: There were no independent experiments performed. On the bases of the exact mass and with the help of MARA the fully epoxidized triglycerides and the ones with one remaining double bond can unambiguously be distinguished, as shown in Fig. 2a (the lines ETG and Unsaturated, respectively).

– Also, you don’t see any addition of MeOH to the epoxide in the ESI-MS? If it is so, this is something you may notice and show a source spectrum in the Supplementary, since it demonstrates that your characterization is solid.

 Reply: No addition of MeOH to the epoxide was observed.

– In fact, in Figure 2 (upper) there are two other rows, one labelled as Unsaturated, the other as Diol, and both are labelled as Filtered (out). From the dot size, I can guess that there is a substantial fraction of unsaturated TGs remaining, and that some has the epoxide ring(s) opened to the diol. Unless this is explained, it is contradictory to the statement of line 131.

 Reply: Indeed, the dot sizes are misleading. The sizes of the Unsaturated and Diol dots are exaggerated for the sake of visibility, they do not reflect the real intensity ratios. It was inserted into the caption of Fig. 2, as follows:

 “The size of the larger dots indicates peak intensity. For the sake of visibility, a minimum dot-size was used, therefore the sizes of the Unsaturated and Diol dots are exaggerated, they do not reflect the real intensity ratios.”

As for the reported MS-MS of isomeric sodiated epoxyTGs, it would be convenient to show the breakdown curves of both isomers, since it is not clear if the two panels of Figure 3 refer to the same isomer or to different ones (I would say they come from different TGs).

 Reply: The breakdown curves of the same peak (m/z 997) of the soybean oil were inserted into the Supplementary, as suggested.

The precursor ions in Fig. 3 are mixtures of epoxyTGs with different fatty acid compositions, for example (18:1), (18:2), (18:3), as the chemical structure shows, or (18:2), (18:2), (18:2). (The overall composition is 57:6.) To make it clear, the following was inserted into the caption of Fig. 3:

 “The chemical structure of the precursor ion (m/z 997) is just an example, it can contain, for example, 3 ELA (18:2) chains, as well.”

Figure S6 mistyped “Collision Energy (eV)”; by the way, it is rather “Collision voltage (DV)”, while center-of mass-collision energy (calculated with the well-known equation Ecm = Elab *(mGas/(mGas + mIon), in eV) is more appropriate when the Authors analyze different precursor ions. Are the curves for the other prominent precursors similar?

 Reply: We modified “Collision Energy” to “Collision Voltage”, as suggested.

An additional breakdown diagram is shown in the Supplementary (Fig. S5b). All the breakdown diagrams were similar to Fig. S5a and S5b.

Thank you for your help!

 Yours sincerely,

 Dr. Sándor Kéki

Professor and Head